# An End-to-End Robust Video Steganography Model Based on a Multi-Scale Neural Network

**Shutong Xu [1], Zhaohong Li [1], Zhenzhen Zhang [2,*] and Junhui Liu [1]**

1   Department of Electronic Information Engineering, Beijing Jiaotong University, Beijing 100044, China
2   Department of Information Engineering, Beijing Institute of Graphic Communication, Beijing 102600, China
*   Correspondence: zhangzhenzhen@bigc.edu.cn

**Abstract:** The purpose of video steganography is to hide messages in the video file and prevent them from being detected, and finally the secret message can be extracted completely at the receiver. In this paper, an end-to-end video steganography based on GAN and multi-scale deep learning network is proposed, which consists of the encoder, decoder and discriminator. However, in the transmission process, videos will inevitably be encoded. Thus, a noise layer is introduced between the encoder and the decoder, which makes the model able to resist popular video compressions. Experimental results show that the proposed end-to-end steganography has achieved high visual quality, large embedding capacity, and strong robustness. Moreover, the proposed method performances better compared to the latest end-to-end video steganography.

**Keywords:** steganography; deep learning; generative adversarial network; robustness

## 1. Introduction

Is it possible that there is a secret sign hiding in a normal-looking picture? The technology of data hiding has the ability to make the answer yes. Data hiding is a technology that aims to embed a piece of information into media cover. What we called steganography is one branch of data hiding, which is dedicated to embedding secret messages into digital multimedia cover such as images and videos, and to ensure that the embedded messages are not perceived or detected by third parties during the transmission process. The characteristic of invisibility makes steganography applicable in many fields. For example, an invisible watermark can be used in copyright protection, or secret military communications. In these fields, steganography has become a very promising technology.

Traditional steganography usually needs to apply relevant expertise to complete the embedding of secret information by modifying the carrier and redundant information in the coding process. However, with the rapid development of deep learning, more and more researchers attempt to apply this technology to steganography. Steganography based on deep learning usually consists of two modules, an encoder and a decoder. The encoder embeds the secret information into carrier and output the steganographic carrier. The decoder is responsible for extracting the secret information from the steganographic carrier. This approach is often referred to as an end-to-end model. On the one hand, compared with traditional methods, an end-to-end steganography model based on deep learning reduces the artificial feature extraction process, and allows neural networks to extract features and complete secret information embedding. On the other hand, the first end-to-end image steganography model called HiDDeN [1] has achieved very high payloads and good visual quality, and the experimental results showed that the end-to-end steganography model based on deep learning can achieve better performance compared with traditional steganography methods.

Although the end-to-end steganography model has obvious advantages, it has not been studied comprehensively. Most of the existing end-to-end steganography algorithms are based on images, for example, Baluja's model [2], Hayes' model [3], SteganoGan [4],

CHAT-GAN [5], and HiDDen [1]. The main structure of these models usually included an encoder, decoder, and steganalyzer which is based on GAN [6]. With the maturity of end-to-end image steganography, its embedding capacity has also increased. From the capacity of 0.203 bits per pixel for Hidden, 0.4 bits per pixel for Hayes', and 4 bits per pixel for SteganoGan, the latest Subramanian [7] propose a large capacity image steganography thatcan embed a colorful image into a cover image of the same size, that is to say, its capacity is 8 bits per pixel.

However, we know that the optional covers of steganography are much more than diverse ever before, including PDF files, pictures posted on social media applications, and videos played on websites. In particular, videos have more redundancy compared with images which is beneficial for hiding more information. However, the research of a video steganography algorithm based on deep learning is inchoate, and the number of relevant studies is very limited. Weng [8] found that hiding highly sparse data is significantly easier than hiding the original frames. Based on this, Wang developed a CNN to hide an inter-frame residual into a cover video. DVMark [9] designed a video steganography network with a differentiable distortion layer to improve robustness, and the steganographic video (abbreviated as stego video) generated by DVMark can resist H.264 [10] compression. RIVAGAN [11] was another end-to-end video steganography model based on GAN, in which the attention module was added to help encoder and decoder extract features better. The stego video generated by RIVAGAN has higher visual quality and higher a extraction rate, but it can only resist one type of video compression which is not the mainstream video compression format. The PyraGAN [12] proposed by Chai introduces coding unit (CU) masks as an input of the encoder which facilitates the network to learn the texture complexity of the video content. However, this method didn't mention the robustness to video compression. Jia [13] also achieved an end-to-end video steganography with robustness to H.264 compression in which the encoder applied both motion synthesis and temporal correlation to maintain the visual quality.

On the other hand, for end-to-end steganography, the way to resist compression is also a key issue. In this regard, existing algorithms propose some solutions. For image compression, HiDDen [1] used the discrete cosine transform (DCT) to build a differentiable compression layer as the approximation of compression algorithms. In order to approximate the rounding calculation which is not differentiable during JPEG quantizing the DCT coefficients, StegaStamp [14] used a piecewise function to replace the quantization steps around zero. Liu [15] introduced a two-stage separable deep learning framework for image watermarking. In the first stage, both encoder and decoder are trained without JPEG compression. In the second stage, the watermarked image will be compressed by real JPEG, and sent to the decoder. Thus, the decoder will learn to how to extract the watermark from a compressed image. This two-stage framework can bypass the non-differentiable process to train the network, but the disadvantage is that only the decoder can learn the features of compression. In order to alleviate the above shortcomings, Jia [16] introduced a noise layer called Mini-Batch of Real and Simulated JPEG compression which had three states including simulated JPEG (similar to HiDDen [1]), real JPEG (to only train the decoder) and identity (no compression). These three states are switched randomly in each batch. Consequently, both encoder and decoder can learn the feature of compression. Unlike the above methods, which want to simulate image compression, RIIS [17] proposed a container enhancement module (CEM) to enhance the stego image, which is utilized as a kind of pre-processing module at the end of the encoder to eliminate the influence of compression distortion like JPEG. For video compression, RIVAGAN [11] built a differentiable layer which is similar to HiDDen's [1] to simulate video compression. Zhou [18] combined 5 kinds of noise including rescaling, median noise, Gaussian noise, weak JPEG compression and strong JPEG compression. Each kind of noise has a trainable weight parameter thatlearns to make the combined noise closer to real video compression distortion. DVMark [9] pre-trained a small 3D-CNN called compression net to mimic the video

compression. To resist distortion of camera photographing and video compression, Jia [13] built a distortion network combined StegaStamp [14] and a 3D light model [19]

All in all, there are many challenges to design an end-to-end video steganography network. First, it requires the steganography algorithm to have a high extraction rate, a large embedding capacity and good visual quality, which strongly depends on an excellent encoding and decoding deep learning network design. Secondly, how to use the temporal continuity feature of video sequences is very critical, as the temporal characteristics are unique for videos compared with images. Moreover, resisting video compression should be considered in the steganography algorithm, as almost all videos would be encoded in one type of coding. However, the compression process is non-differentiable, so it cannot be directly put into the deep learning network; consequently, the way how to make the network learn the compression process is also one of the key issues.

In this paper, we propose a video steganography algorithm based on end-to-end multi-scale network which can resist kinds of video compression formats. First, we choose video as a carrier in this algorithm, not only because video as a digital medium is spread widely, but also because video has a large amount of information and a large amount of information redundancy, which is suitable to become a steganographic carrier. Secondly, in order to improve the feature learning of the network, we use GAN as basic network design, which contains the encoder, decoder, and discriminator. The encoder-decoder and the discriminator are in a competitive relationship, and the discriminator is helpful to improve the algorithm's ability to resist steganalysis. Also, we added a pyramid-like network design to the encoder and decoder. The pyramid-like network architecture with up and down sampling processes can significantly improve the feature extraction ability of the network. In addition, in order to effectively extract and utilize the temporal features of the video, most of the convolutional layers in the network apply 3D convolution. As illustrated before, how to resist compression is an important challenge for video steganography. For this, we introduce a noise layer between the encoder and the decoder (because we consider the changes generated by video compression as noise) and then add it into the network. Based on the above network design, the experimental results show that the proposed end-to-end video steganography has achieved a very high extraction rate, a large embedding capacity, and good visual quality. Furthermore, it finally can resist some popular video compressions.

This paper is organized as follows: Section 2 introduces the design of the end-to-end steganography network including the encoding network, the decoding network, the discrimination network, and the noise layer. Section 3 shows all the experimental results. The conclusion is summarized in Section 4.

## 2. Proposed End-to-End Video Steganography Network

### 2.1. The Architecture of the End-to-End Video Steganography Model

In this paper, we propose an end-to-end video steganography algorithm which aims to hide n-bit binary messages in a cover video. The architecture of the proposed method is shown in Figure 1 which consists of four main components: (1) an Encoder **E** takes binary message ($M_i$) and a cover video ($V_{cover}$) as input, and outputs stego video ($V_{stego}$); (2) a Noise layer **N** is mainly responsible for adding the process of video compression as noise to the stego video; (3) a Decoder **D** receives the noised stego video ($V'_{stego}$) and extracts the hidden message ($M_o$) from it; (4) a Discriminator **Ds** is used to distinguish the cover video ($V_{cover}$) and the steganographic video ($V_{stego}$) and outputs the judgment scores ($S_{cover}$ and $S_{stego}$). In addition, there are a total of 3 loss functions to constrain the network, making the encoder-decoder and the discriminator in a state of competition, thus forming the entire GAN. More detailed design will be introduced below.

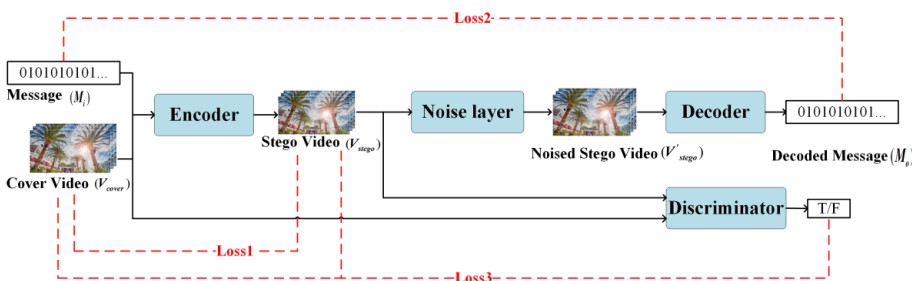

**Figure 1.** Network architecture.

2.1.1. Encoder Network Design

The main function of the encoder is to embed messages into the cover video and generate a stego video, so the video feature extraction ability and message embedding ability of the encoder are very important. The encoder network is designed as shown in Figure 2, in which cover video ($V_{cover}$) and message ($M_i$) are the input, and stego video ($V_{stego}$) is the output shown as Equation (1):

$$V_{stego} = E(V_{cover}, M_i) \tag{1}$$

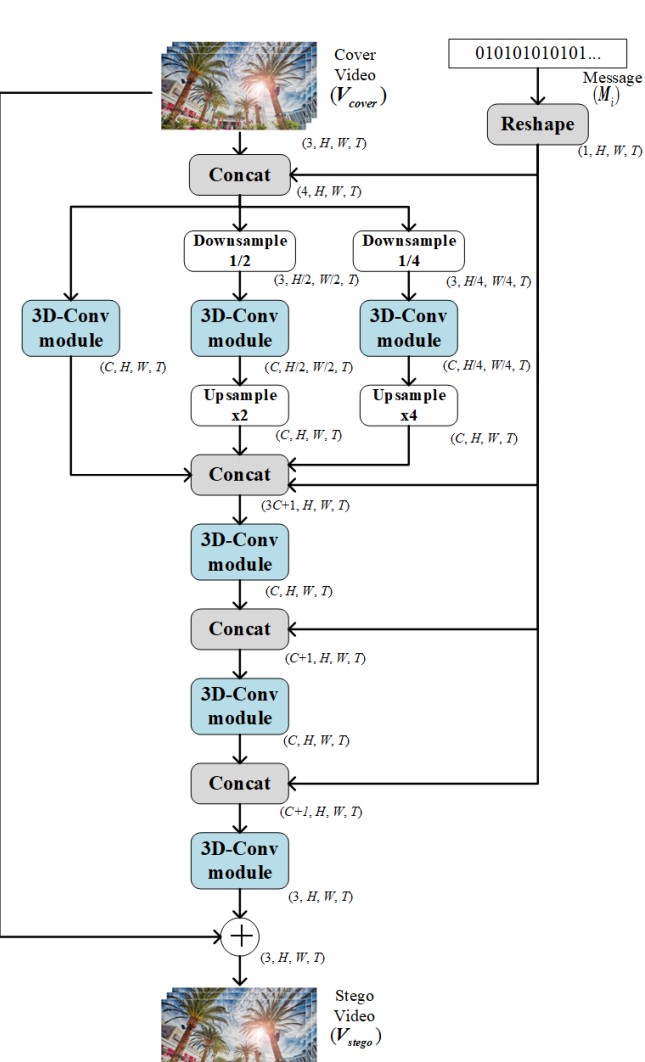

**Figure 2.** The network design of encoder.

In the Encoder network, labels such as ($C, H, W, T$) refer to the size of each output, which are channel, height, width, and frame, respectively. The cover video usually contains

hundreds or thousands of frames with different resolutions. To unify the input to the Encoder, the resolution of the cover video is cropped to $H \times W$, and every $T$ continuous frames are taken as a unit. Such a cover video unit will be used as input to the Encoder. The uniform input size facilitates the feature learning of the neural network, and the fixed number of frames also makes the extraction of temporal features more efficient. Note that the cover video in the following paper generally refers to a video unit of size ($C$, $H$, $W$, $T$).

First, reshape the message to the size of (1, $H$, $W$, $T$) so that it can be concatenated with the cover video, the size of which is (3, $H$, $W$, $T$) to form a feature data of (4, $H$, $W$, $T$). Then we introduce a multi-scale convolution network, in which the 4-channel features will be made into 3 copies and entered into three different convolution paths. There are 3D-convolution modules in each path including the 3D-Conv layer, LeakyReLU [20], and Batch normalization [21]. In general, neural networks usually use the 2D-Conv layer, which is suitable for image. However, for video, itis more suitable to apply 3D-Conv layer to extract temporal features. In two of the paths among multi-scale networks, the features are first downsampled to one-half and one-quarter scale, then passed through the 3D-Conv module, and finally upsampled to their original size, so that the three-scale features can be concatenated with the reshaped message again. The multi-scale network designed above can make the encoder more effective in extracting features. For example, the original scale path can extract small-scale features, and the half-scale and quarter-scale can expand the receptive field of the convolution kernel to make features more macroscopic. In addition, the convolution module of the entire encoder uses 3D-Conv, which can better learn the temporal feature of video than two-dimensional convolution. After that, inspired by DenseNet [22], the feature will be repeatedly convolved and concatenated with the reshaped message. Such a network connection can learn features of the message multiple times, and increase the weight of the message in the network to ensure that the message is successfully embedded into the cover video. At the end of the encoder, the cover video will be added to the learned feature of (3, $H$, $W$, $T$), and the stego video will be output. Therefore, in fact, the encoder learns the residual between the cover video and the stego video, and the steganographic video can be obtained by adding the residual containing the message feature and the cover video. This idea from ResNet [23] can effectively improve the image generation ability of the network, avoid gradient disappearance or gradient explosion, and finally improve the visual quality of the stego video.

### 2.1.2. Decoder Network Design

The decoder receives the noised stego video ($V'_{stego}$) and extracts the decoded message ($M_o$) from it shown as Equation (2), so functionally this is somewhat the opposite of the encoder. Therefore, the design of the decoder is similar to that of the encoder, which is also composed of the 3D-Conv module and multi-scale networks (as shown in Figure 3). The 3D-Conv module helps the decoder to extract the temporal features of the stego video, and the multi-scale network is helpful to expand the receptive field and improve the message extraction rate We have

$$M_o = D(V'_{stego}, M_o) \qquad (2)$$

First, the stego video will go through three convolution paths of different scales, then concatenate three features of size ($C$, $H$, $W$, $T$) to get a feature of size ($3C$, $H$, $W$, $T$). Finally, the feature is sent to the 3D-Conv module which outputs a message of size (1, $H$, $W$, $T$). Note that all the 3D-Conv module in decoder is the same as that in the encoder.

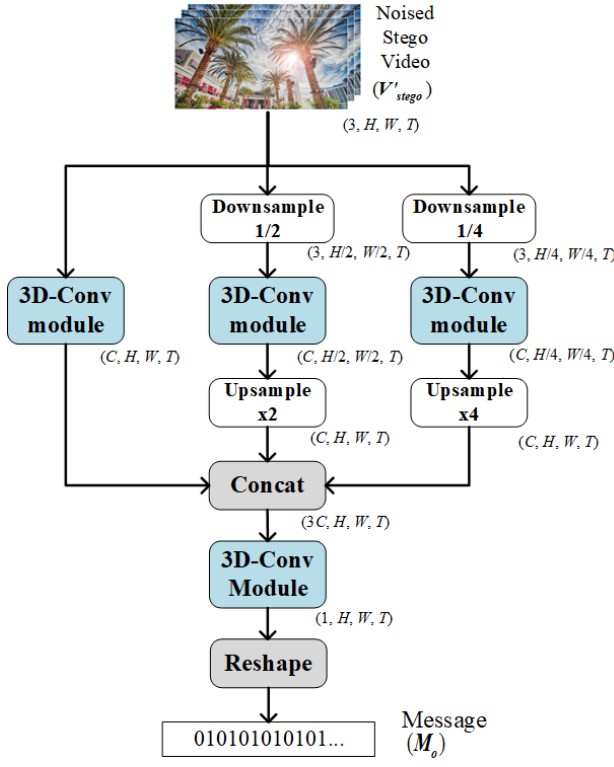

**Figure 3.** The network design of decoder.

### 2.1.3. Discriminator Network Design

The discriminator takes cover video ($V_{cover}$) and stego video ($V_{stego}$) as input, and finally outputs the judgement scores ($S_{cover}$ *and* $S_{stego}$) shown as Equation (3). Because the discriminator only needs to extract the features of the input video and output the judgment results, it does not need a complicated network. As shown in Figure 4, after the input video passes through four 3D-Conv modules, a feature map of size (1, $H$, $W$, $T$) is obtained, and then the average value of the feature is calculated to get a value ranging from 0 to 1. Greater than 0.5 means that this video is judged to be a stego video, otherwise this video is judged to be a cover video. We have

$$\left(S_{cover}, S_{stego}\right) = Ds\left(V_{cover}, V_{stego}\right) \tag{3}$$

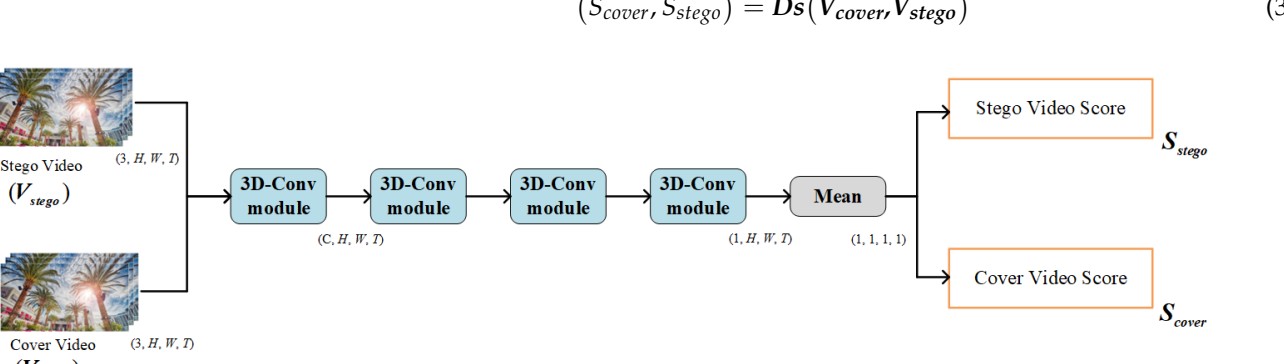

**Figure 4.** The network design of discriminator.

### 2.1.4. Noise Layer Design

Before the video is transmitted, it is usually encoded by an encoder which commonly uses H.264 or H.265 (HEVC) [24] as an encoding standard. HEVC applies a more flexible partition structure and a more accurate prediction algorithm based on the H.264 coding framework. The brief encoding diagram of HEVC is shown in Figure 5. The input video is

first divided into image block units of different sizes, and then the intra-frame prediction or inter-frame prediction is performed on each unit. The prediction block will be subtracted from the original block to obtain the residual, which will be further transformed and quantized. Finally, entropy encoding is performed on the quantized transform coefficients to output the bitstream. The quantization process (marked in red in Figure 5) is non-differentiable and will cause information loss, so handling the transformation and quantization process is the key of improving the anti-compression ability for the end-to-end steganography model.

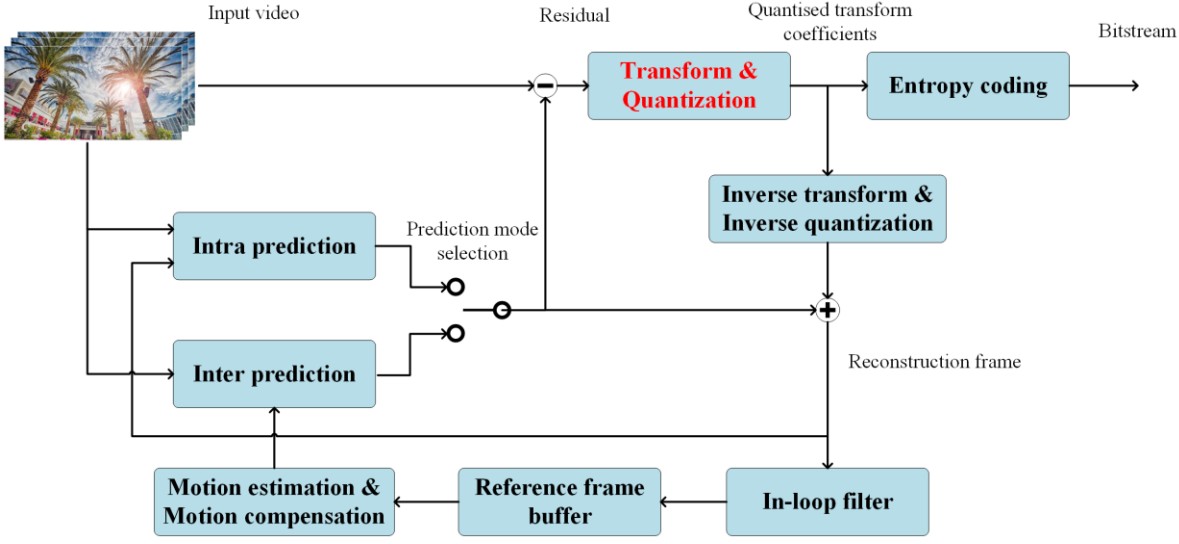

**Figure 5.** Simplified block diagram of HEVC encoder.

As mentioned, the quantization in the video encoding process involves rounding operations which is non-differentiable. This means that video compression cannot be directly incorporated into the neural network based on gradient back-propagation [25]. Inspired by Zhang [26], we treat the distortion caused by video encoding as noise and add it to the stego video. Note that general noise can be directly added to video in the neural network and this operation is differentiable. The flow chart of noise layer is shown in Figure 6.

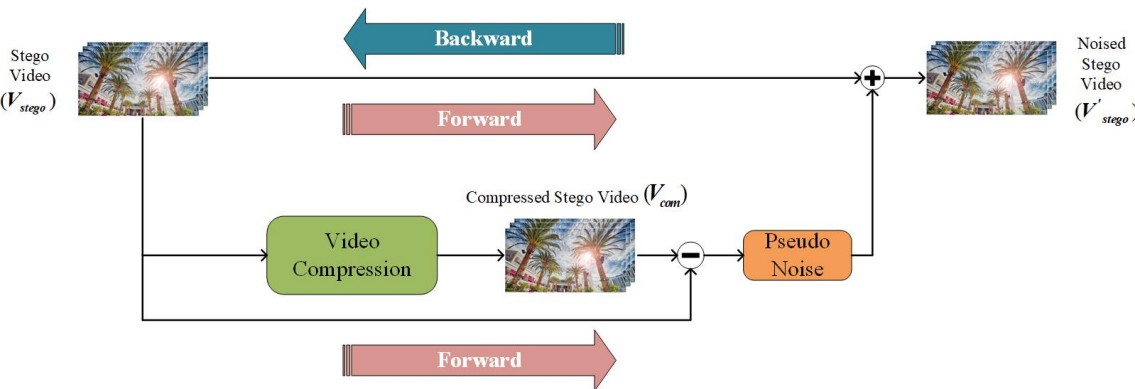

**Figure 6.** The flow chart of noise layer.

There are two pathways in the Figure 6, the upper path performs forward-propagation and back-propagation, and the lower path only has forward-propagation which means there is no need to calculate and store the gradient of variables. In the lower path, stego video ($V_{stego}$) goes through video compression process to get compressed stego video ($V_{com}$). Then it will subtract the stego video ($V_{stego}$) to get the distortion caused by the

compression. The distortion without gradient is regarded as a pseudo noise $\varepsilon = V_{com} - V_{stego}$ which is then added to the stego video ($V_{stego}$) to get a noised stego video ($V'_{stego}$) shown as Equation (4). The role of such noise layer is to convert video compression into noise generation, and make the neural network robust against such pseudo noise. We have

$$V'_{stego} = N(V_{stego}) = V_{stego} + \varepsilon \tag{4}$$

### 2.2. Loss Function

As shown by the red dotted line in Figure 2, there are three loss functions in the network framework, Loss1 is for Encoder, Loss2 is for Decoder, and Loss3 is for Discriminator. After multiple rounds of back-propagation and model parameter updating, the loss functions are minimized, and finally we can obtain a model with better performance.

1.  Loss1 ($L_1$) which constrains the encoder uses mean square error (MSE) as Equation (5). The purpose of $L_1$ is to make the stego video $V_{stego}$ as similar to the cover video $V_{cover}$ as possible. Note that the loss function compares two videos, so the number of frames T will be calculated in the formula. We have

$$L_1 = MSE\left(V_{cover}, V_{stego}\right) = \frac{1}{C \times H \times W \times T}\|V_{cover} - V_{stego}\|_2^2 \tag{5}$$

2.  Loss2 ($L_2$) uses cross entropy loss to measure the message extraction ability of the decoder. The more consistent the input message $M_i$ and decoded message $M_o$ are, the smaller the $L_2$. We have

$$L_2 = CrossEntropy(M_i, M_o) \tag{6}$$

3.  In Equation (7), $S_{cover}$ and $S_{stego}$ are judgment results from the discriminator for cover video $V_{cover}$ and stego video $V_{stego}$ respectively. The range of $S_{cover}$ and $S_{stego}$ is 0 to 1. If it is greater than 0.5, it means that this video is judged to be a stego video, otherwise this video is judged to be a cover video. Therefore, the smaller the $L_3$, the stronger the steganalysis ability of the discriminator. We have

$$L_3 = S_{cover} - S_{stego} \tag{7}$$

During the training process, the encoder and decoder are concatenated together to form the encoder-decoder. The gradient backpropagation and model parameter update for encoder-decoder are performed simultaneously, so a joint loss function $L_{enc-dec}$ shown in Equation (8) is used for the whole network. The discriminator is trained separately, by using $L_{dis}$ shown in Equation (9), where $\lambda_1, \lambda_2, \lambda_3, \lambda_4$ are weight factors, which will change with the experimental configuration. We have

$$L_{enc-dec} = \lambda_1 L_1 + \lambda_2 L_2 + \lambda_3 S_{stego} \tag{8}$$

$$L_{dis} = \lambda_4 L_3 \tag{9}$$

Note that Equation (8) also contains $S_{stego}$, which is expected to be smaller because of the need for minimization, which is contrary to the minimization direction of $L_{dis}$. Specifically, during the training process of encoder-decoder, it needs to reduce the value of $S_{stego}$, that is, the stego video $V_{stego}$ needs to be similar to the cover video $V_{cover}$. However, during the training process of the discriminator, it needs to distinguish stego video $V_{stego}$ and the cover video $V_{cover}$ as much as possible, that is, $S_{cover}$ approaches 0, and $S_{stego}$ approaches 1. The design of such loss functions make the encoder-decoder network competes with Discriminator network to form a GAN, which can make the whole network perform better. In the following experiments, because the primary network is Encoder-Decoder, the weight factors are adjusted to make the parameter updates of the GAN more biased towards the encoder-decoder network.

All the proposed design of network above including the encoder, decoder, discriminator, noise layer, and loss functions form the end-to-end video steganography model.

## 3. Experimental Results and Analysis

For software and hardware, we train and evaluate the proposed steganography network based on a GeForce RTX 2080Ti with driver version of 430.64, and the CUDA version is 10.1. In terms of the hyperparameters of the model, Adam is chosen as the optimizer with learning rate 0.0001, and the weight decay is 0. The number of epochs is 96. The batch size is 16. Other hyperparameters are set as default. The experimental data below are all based on the Vimeo-90k [27] data set. For a HEVC compression experiment, we randomly select 800 video clips for training, and 100 video clips for validating. The video clips are cropped to a size of $5 \times 64 \times 64$. Thus, the size of cover video is $64 \times 64$ with 5 continuous frames, as 5 frames usually can guarantee continuity and similarity of temporal features.

### 3.1. Evaluation Indicators

For video steganography algorithms, stego video visual quality, message embedding capacity, message extraction accuracy and robustness to video compression are important evaluation indicators. First, we use peak signal to noise ratio (PSNR) as a measure of video quality, which is shown in Equation (10). MSE can be calculated according to Equation (1). Furthermore, $A$ represents the peak signal, here with the value of 255. We have

$$PSNR = 20 \times log_{10}(\frac{A}{\sqrt{MSE}}) \tag{10}$$

For message embedding capacity, bits per pixel (BPP) is used to measure how many bits can be embedded in the cover video shown in Equation (11), where $L$ represents the length of message, and $H \times W \times T$ represents the size of cover video. We have

$$BPP = \frac{L}{(H \times W \times T)} \tag{11}$$

At last, we use a Accuracy to calculate the consistency of the input message and decoded message as a measure of message extraction ability and robustness to video compression shown in Equation (12), where $L$ represents the length of message, $M_i^k$ and $M_o^k$ represents the k-th binary bit of input message and decoded message, respectively. We have

$$Accuracy = \frac{\sum_{k=1}^{L}\left(1 - \left|M_i^k - M_o^k\right|\right)}{L} \tag{12}$$

### 3.2. Performance of the Proposed Model

This section mainly introduces two parts of the experiment. The first is the network performance without the noise layer to verify the feasibility of the proposed network. After that, it will show the experimental results of the network added to the noise layer to verify its resistance to video compression.

#### 3.2.1. Experimental Results without Noise Layer

In this section, we remove the noise layer, and only kept the encoder, decoder and discriminator to verify the feasibility of the model design without considering resistance to video compression, whereas the testing of the proposed network with noise layer and the robustness to video compression will be presented in the next section.

First, the cover video is cropped to a size of $5 \times 64 \times 64$, and then the embedding capacity is set to be 1.0 BPP, that is, each pixel will be embedded with 1 bit. Thus, a cover video with a size of $5 \times 64 \times 64$ will be embedded with a total of $5 \times 64 \times 64$ bits. The binary message is randomly generated by the code. This capacity is much larger than

some other video steganography, like 96 bits in 8 × 128 × 128 for DVMark [9], 100 bits in 400 × 400 for Jia's [13] and 256 bits in 8 × 128 × 128 for RIVAGAN [11].

The experimental results of visual quality and extraction accuracy are shown in Table 1. Without noise layer, it can be seen that the PSNR of the stego video reaches 43dB, which means it has a very high visual quality. It also can be seen from the visual display in the second row of Figure 7 that there is almost no difference between the stego video and the cover video. It is hard to distinguish the cover video and the stego video with the naked eyes.

**Table 1.** Performance of model without noise layer.

| Method | Cover Size | BPP | PSNR (dB) | Accuracy |
|--------|-----------|-----|-----------|----------|
| Ours | 5 × 64 × 64 | 1.0 | 43.036 | 1.000 |
| PyraGAN [12] | 256 × 256 | 1.0 | 43.109 | 0.994 |

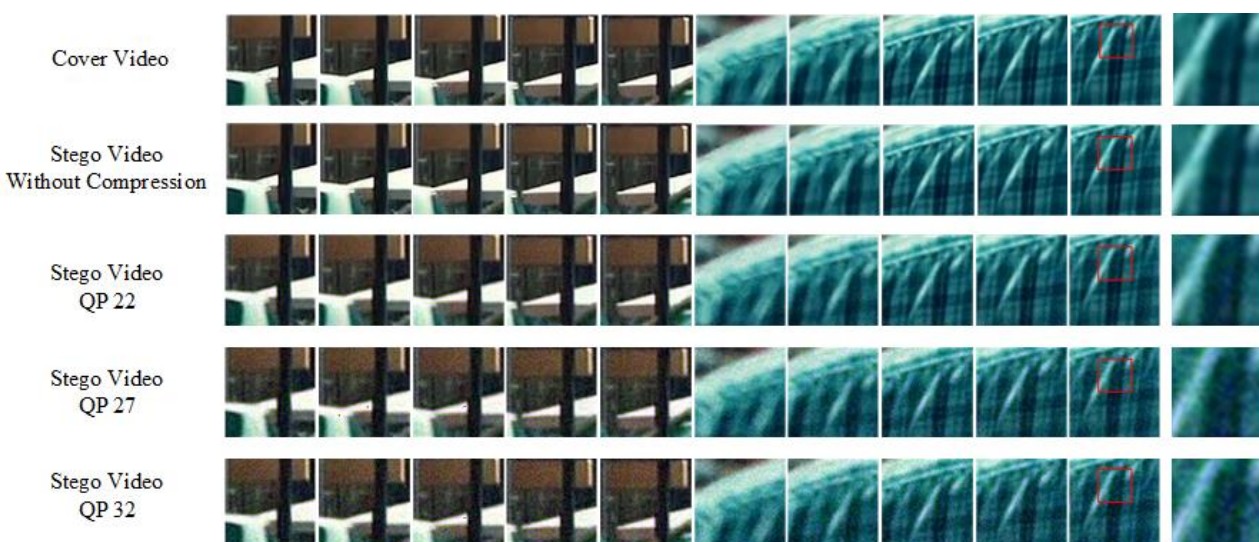

**Figure 7.** Cover video and stego videos.

On the other hand, the accuracy of the secret message extraction is close to 1, which means the extraction message is almost the same as the embedded secret message. Note that the Accuracy in Table 1 is a result of rounding, so there are still very few bit errors. As the high visual quality of the stego video and the ideal extraction accuracy, the proposed method can be used as a method of secure and effective secret communication.

Furthermore, we compare the proposed model without the noise layer to the method of PyraGAN [12], the reason is that PyraGAN [12] is a new published video end-to-end steganography in 2022, and it didn't consider the resistance to video compression, so it is suitable to compare it with the proposed model without the noise layer in capacity, visual quality and accuracy except the robustness. Table 1 shows the performance of PyraGAN [12], the capacity of it is the same as ours which is 1.0 BPP The proposed method without a noise layer gets a little bit better performance in extraction accuracy and similar visual quality (PSNR). However, in this way, the stego videos produced by the proposed method without noise layer and the method of PyraGAN [12] are supposed to be very fragile with regard to video compressions, and this assumption will be tested and discussed in Section 3.2.2.

### 3.2.2. Experimental Results with Noise Layer

First, the resistance to video compression of the proposed method without noise laryer and the method of PyraGAN [12] is tested. HEVC is chosen for compression because it is

the next-generation video coding standard for H.264, which is also widely used in social media. The cover video size and BPP are the same as above. In the testing, the stego video produced by the proposed method without noise laryer and the method of PyraGAN [12] will directly undergo HEVC compression with different quantization parameters (QPs). A quantization parameter (QP) represents the level of compression. The smaller the QP, the smaller the quantization step, and the smaller the distortion caused by compression.

As shown in Table 2, it can be seen that after compression the extraction accuracies of the proposed method without noise layer and the method of PyraGAN [12] drop a great deal compared with those before compression in Table 1, most of the accuracies are close to 50%, which proves the proposed method without noise layer and the method of PyraGAN [12] fail to resist to video compression.

**Table 2.** Performance comparison of different QPs.

| QP | Method | PSNR (dB) (before Compression) | PSNR (dB) (after Compression) | Accuracy |
|---|---|---|---|---|
| | Ours | 38.493 | 31.894 | 0.976 |
| 22 | Ours (no noise layer) | 43.222 | 38.685 | 0.616 |
| | PyraGAN [12] | 42.419 | 39.200 | 0.514 |
| | Ours | 35.777 | 29.026 | 0.968 |
| 27 | Ours (no noise layer) | 43.097 | 38.870 | 0.520 |
| | PyraGAN [12] | 42.419 | 36.356 | 0.504 |
| | Ours | 32.473 | 27.029 | 0.795 |
| 32 | Ours (no noise layer) | 43.036 | 37.010 | 0.520 |
| | PyraGAN [12] | 42.419 | 33.189 | 0.501 |

However, video compression is the most popular operation for videos, so the resistance to video compression is an important issue in the end-to-end video steganography. The difficulty is how to simulate video compression in the deep learning network because the processing of video compression is complicate and undifferentiable. Existing end-to-end video steganography almost used a differentiable layer or a small network to simulate the video compression, but none of them could mimic it perfectly. Consideringthis, we would like to use a better and general method to resist video compression, and the result is using the real compression noise in the noise layer as introduced in Section 2.1.4. The next issue is how to measure the robustness to video compression. As we know, the distortion brought by compression will directly lead to the decline of the visual quality, and the secret message hiding in the stego video is easily destroyed, which makes it hard for the decoder to extract the secret message correctly. Therefore, visual quality of the stego video after compression and the extraction accuracy of the secret message are usually used to reflect the robustness to video compression in video steganography, and the weaker the robustness, the worse the visual quality, and the smaller the accuracy.

In the proposed model, the noise layer using real compression noise between the encoder and the decoder is a key design for resisting compressions. That is to say, after the noise layer is added, the encoder and decoder can learn the feature of video compression. The encoder will raise the weight of embedded information by sacrificing certain visual quality, and the decoder will learn to extract the message from a compressed stego video. During training process of the proposed model, the noise layer is added between the encoder and decoder which includes HEVC compression. During the validation process, the stego video will directly undergo HEVC compression. We can see from Table 2 that by adding the noise layer, all the extraction accuracies of the decoded message can reach greater than 80%, especially when the compression is not very serious, it can reach 96% or more. Facing the distortion of video compression, the stego video maintains good visual quality (PSNR) and prevents the drop of Accuracy which proves the strong robustness of the proposed model.

Figure 7 further shows the visual quality of cover video and stego video. The first row is two cover videos with 5 continuous frames respectively. The second row includes stego video frames from the proposed model without noise layer. The third row includesthe stego video frames from the proposed model adding noise layer of HEVC with QP 22. The fourth and fifth row are similar to those above except for QP 27 and QP 32. The last one of each row is an enlarged image of the red box in the last frame to clearly compare the differences in visual details. It can be seen that there will be some noise in the stego video, especially in enlarged images. The higher the QP, the more obvious the noise. In general, the visual quality is within an acceptable range.

All in all, by adding the noise layer, the ability of the algorithm to resist video compression is achieved, and the good visual quality and high accuracy can still be maintained.

### 3.3. Comparison Experiment

In this section, we compare several end-to-end video steganography networks that are also resistant to video compression. For a comparison experiment, we randomly select 1264 video clips for training, and 128 video clips for validating. Video clips are cropped to a size of $8 \times 128 \times 128$ which is the same size as comparison methods for fairness of comparison. Thus, the size of cover video for comparison experiment is $128 \times 128$ with 8 continuous frames.

As shown in Figure 3, the secret message needs to be reshaped to the same size as the stego video in order to be concatenated with the stego video. However, because the BPP of each algorithm is different, for fair comparison, we reduce the message length to achieve the same or similar BPP as that of the comparison algorithms. For example, in Figure 8, on the left is the input message, which is first reshaped and arranged into a matrix. Then, each bit in the matrix will be repeated $x^2$ times, and if necessary, they will also be repeated in the time domain. At last, the repeated message matrix will be the same size as the stego video, and the number of $x$ is determined by the ratio of the video size to the message length. If $x$ is not an integer, the repeated message matrix will be padding with 0.

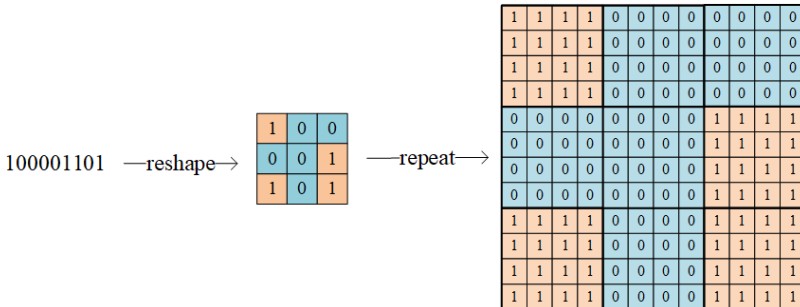

**Figure 8.** An example of expanding message.

We will compare with the methods of DVMark [9], Jia's [13], and RIVAGAN [11]. These algorithms are the latest end-to-end video steganography, and all of them are resistant to video compression. Because the experimental configuration of these is not the same, some adjustments will be made as in the following.

First, the cover video size of DVMark [9] is $8 \times 128 \times 128$, a total of 96 bits can be embedded, and it is resistant to H.264 (CRF = 22), where CRF is a parament of compression level. CRF is usually set in a range of 18 to 28, and the larger the CRF, the greater the compression rate. When CRF is 22, the compression rate will be above 50%. In the comparison experiment, the cover video in the proposed method will be cropped to the same size, and the video compression process of the noise layer will be changed to H.264 (CRF = 22). By the expanding method mentioned in Figure 8, it is hard to change the length of the message to 96 bits. As a result, the message is adjusted to 100 bits, which is a bit more than 96 bits in DVMark. In general, in terms of a video steganography, more embedded message usually means there is more distortion to the stego video which also leads to the

decline of PSNR and Accuracy. However, it can be seen from Table 3 that the proposed steganography gets much better PSNR and higher extraction accuracy than DVMark even with a message with a longer length, which illustrates the proposed method outperms the method of DVMark in all of these three aspects.

**Table 3.** Comparison results with DVMark [9], Jia's method [13] and RIVAGAN [11].

| Compression Standard | Method | Cover Size | Length of Message (Bits) | BPP | PSNR (dB) (before Compression) | Accuracy |
|---|---|---|---|---|---|---|
| H.264 (CRF = 22) | DVMark [9] | 8 × 128 × 128 | 96 | 0.000732 | 36.50 | 0.980 |
|  | Ours | 8 × 128 × 128 | 100 | 0.000763 | 42.25 | 0.990 |
| H.264 (CRF = 23) | Jia's [13] | 400 × 400 | 100 | 0.000625 | 40.28 | 0.977 |
|  | Ours | 5 × 400 × 400 | 5×100 | 0.000625 | 41.10 | 0.998 |
| MJPEG | RIVAGAN [11] | 8 × 128 × 128 | 256 | 0.001953 | 42.05 | 0.992 |
|  | Ours | 8 × 128 × 128 | 256 | 0.001963 | 45.19 | 0.994 |

Next, the cover video size of Jia [13] is 400 × 400, each frame of Jia's stego video can embed 100 bits, and it is also resistant to H.264 but the CRF = 23. Considering the GPU memory, we select our cover video size as 5 × 400 × 400, and embedded a total of 5 × 100 bits to make BPP equal to the front. The video compression process of the noise layer will be H.264 (CRF = 23). Table 3 shows that the proposed model surpasses Jia's in both PSNR and Accuracy.

At last, the cover video size of RIVAGAN [11] is also 8 × 128 × 128, a total of 256 bits can be embedded, and it is resistant to MJPEG [28]. Thus, in the comparison experiments with the method of RIVAGAN, the cover video and the length of message in the proposed method will be adjusted to the same size, and the video compression process of the noise layer will be changed to MJPEG. OpenCV is used to compress the video, and the paraments of MJPEG are set as default, which is the same as RIVAGAN. Table 3 shows that the proposed steganography achieved a higher visual quality and message extraction accuracy than RIVAGAN.

As mentioned in the Introduction, DVMark [9] used a pre-trained 3D-CNN to mimic video compression. Jia [13] combined StegaStamp [4] and light model [19], and RIVAGAN [11] used a differentiable noise layer of DCT to simulate video compression. Essentially, these methods let the model learn the feature of 3D-CNN or differential distortion layer but real video compression. It is hard to design a 3D-CNN or distortion layer to simulate video compression perfectly. However, the pseudo noise in our noise layer comes from the real video compression. That may help our model to resist video compression and perform better than DVMark [9], Jia's method [13] and RIVAGAN [11].

Figure 9 shows the visual quality of the cover video and stego video. The first row is a cover video with 8 continuous frames. The second row includes stego video frames from the proposed model with the noise layer of H.264 (CRF = 22). The third row includes also stego video frames from the proposed model, but with the noise layer of MJPEG. The last one of in each row is an enlarged view of the red box in the last frame. Except for some slight subtle texture differences, the difference is nearly invisible to the naked eye. From the experimental results, it can be seen that our method can still resist H.264 and MJPEG, and surpasses other algorithms in visual quality (PSNR) and extraction rate (Accuracy) which proves the better robustness to video compression.

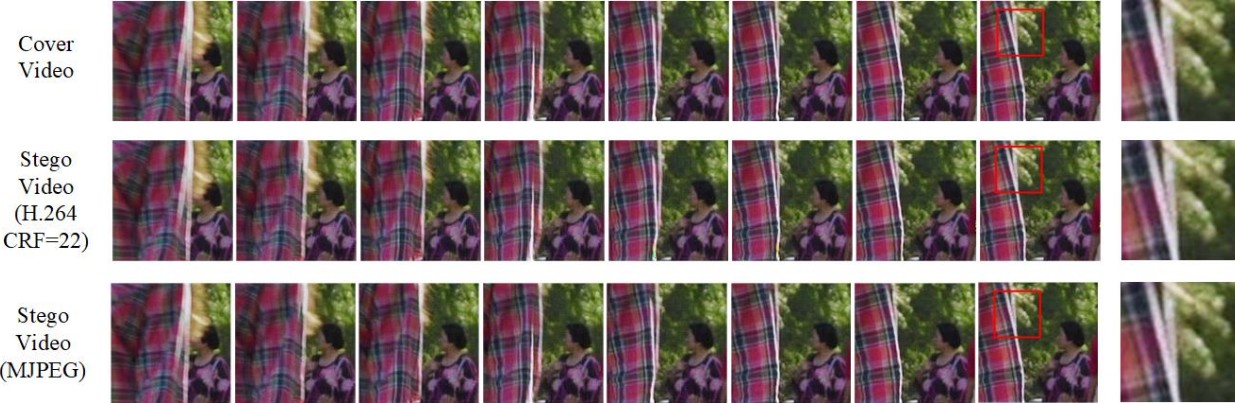

**Figure 9.** Cover video and stego videos (H.264 CRF = 22 and MJPEG mean the stego video will be compressed by H.264 with CRF = 22 and MJPEG in the noise layer respectively).

## 4. Conclusions

In this paper, an end-to-end video steganography for transmitting secret information is proposed. The proposed model consists of encoder, decoder and discriminator, and a noise layer which uses real compression noise is introduced between the encoder and decoder to achieve strong resistance to video compressions of HEVC, H.264 and MJPEG. In the proposed steganography, continuous video frames and secret message are sent to the encoder as the input, and 3D convolutional layers are appled to better utilize the temporal characteristics of videos and multi-scale networks to extract features of different scales. Experimental results show that the proposed steganography not only has very high visual quality and decoded message accuracy, but also performs better in video compression robustness compared with the state-of-the-art work. In the future, the research will focus on designing a novel noise layer for better simulating video compressions.

**Author Contributions:** Conceptualization, S.X. and Z.L.; methodology, Z.Z.; software, S.X. and J.L.; experiment, S.X., J.L.; writing—original draft preparation, S.X.; writing—review and editing, Z.L. and Z.Z.; funding acquisition, Z.Z. All authors have read and agreed to the published version of the manuscript.

**Funding:** This research was funded by The Scientific Research Common Program of Beijing Municipal Commission of Education, grant number: KM202110015004.

**Institutional Review Board Statement:** Not applicable.

**Informed Consent Statement:** Not applicable.

**Data Availability Statement:** Not applicable.

**Conflicts of Interest:** The authors declare no conflict of interest.

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
