# Peer review of "An End-to-End Robust Video Steganography Model Based on a Multi-Scale Neural Network"

_electronics, doi:10.3390/electronics11244102_

Round 1

Reviewer 1 Report

Based on my review:

1. Low contributions based on the current article contents. Unclear framework and please provide the mathematical functions on each stage.

2. No clear results and discussion on the high visual quality, large embedding capacity and strong robustness.

3. Please do some comparisons with the state-of-the-arts methods.

4. No clear proof and discussion on the stego video can resist video compressions, please provide the relationship with  the proposed framework.

5. Poor presentation on the experimental results that the proposed model has very high visual quality and extraction accuracy without noise layer.

Author Response

Thanks a lot for your comments.  About Our respones, please see the attachment. We have rewritten some contents in revised manuscript.

Reviewer 2 Report

Steganography is the practice of hiding information inside something that appears normal and is not secret. Steganography technology is significant to the attack and defense of information. In this study, the authors proposed a robust end-end video steganography model to hide n-bits messages in a video. However, there are some issues that need to be addressed before publication. My comments and suggestions are as follows:

1. In Figure 3, before the output of the decoder, a reshape module is needed to add to convert (1, H, W, T) to n-bits messages.

2. In Figure 4, after the Mean module, is the feature map size (1,1,1,1)? 

3. In Page 6, "The brief encoding diagram of HEVC is shown in Figure 1." should be "The brief encoding diagram of HEVC is shown in Figure 5." 

4. Equations 7 and 8 are the same.

5. Please compare existing work with the proposed method without the noise layer.

6. In Page 11, "But for the convenience of repetition, the message will be adjusted to 100 bits." is not convincing. A fair comparison should be under the same setting.

7. In Tables 3 and 4, please add the PSNR values after compression.

8. What are QP values when you implement the proposed method to compare with DVMark and RIVAGAN algorithms?

9. The robustness of the designed algorithm has not been shown, please consider showing more experimental results.

Author Response

(The authors gave the same response as above.)

Reviewer 3 Report

The authors propose an end-to-end video steganography algorithm based on GANs and multi-scale deep learning networks. The related works investigated in this paper are few and the experiments are insufficient. I think the quality of the article still needs to be improved. The specific comments are as follows:

 Introduction

1. There are few recent related works mentioned in the introduction. I think the related work and corresponding descriptions in the introduction need to be expanded. The descriptions of related work in this section are too brief and do not reflect the issues raised by the authors. As described in the fifth paragraph of the introduction, the challenges faced by end-to-end video steganography networks should be included in the corresponding work and described in detail (at least reflecting the problems that exist).

2. Furthermore, I don't think a description of the experimental results should appear in the introduction. The introduction part is to explore and analyze the research status, existing problems and the rationality of the methods to be proposed for the relevant problems.

 Experiment

1. The description of the experimental details is incomplete. The learning rate, decay rate, optimizer, and hyperparameter settings for network training are not specified.

2. Equation 7 and Equation 8 are repeated, and there is no formula description for the calculation of the accuracy rate.

3. The experimental setup is chaotic. Table 3 and Table 4 have the same test content and can be combined into one table. The experimental comparison algorithm is not convincing enough to only compare two algorithms. Although the two algorithms mentioned in the article are recent related work, methods without compression resistance can also be compared, which can better illustrate the advantages of the proposed method. In particular, the structure is similar to the recently published work https://doi.org/10.3390/electronics11071142 in the journal Electronics, which is not mentioned in the introduction and is not described in the experimental section.

4. Figure 7 and Figure 9 in the experimental section suggest adding auxiliary markers to highlight the advantages of the proposed method.

5. The author describes the advantages and design features of the proposed network in detail in the methods section. I think it is appropriate to add specific training details in the experimental section, which is conducive to echoing the content mentioned in the methods section.

 Conclusion

The content description of the conclusion part is poor.

1. The conclusion part should condense the proposed method to highlight the key points of the work, rather than simply repeating the description.

2. The content description is not concise. “Experimental results show that” and “It also can be seen from the comparison experiments” appear at the same time in the conclusion section, which may reflect that the author's summary of the proposed work is imperfect.

3. When looking forward to future work, I think it is appropriate not to describe statements such as “such as”.

 References

From the number of references, it can be seen that the author's research on related work is not enough. It is recommended to cite more papers that have been officially published in recent years to improve the persuasiveness and novelty of the paper.

Author Response

(The authors gave the same response as above.)

Round 2

Reviewer 1 Report

Based on the updated article:

1. The authors were unable to answer question No 2.

2. No clear discussion on the robustness of the proposed model. The contents must related to the challenges.

3. Please provide more methods for comparison.

Author Response

(The authors gave the same response as above.)

Reviewer 2 Report

The authors have addressed all my comments. I don't have any further questions about this paper.

Author Response

Thank you a lot. There are some changes in revised manuscript according to other reviewers.

Reviewer 3 Report

1In my opinion, the author needs to moderately increase the relevant references to enrich the scientific rationality of the article.

2The conclusion needs to be improved. I do not think it is appropriate for a description of the status quo to appear in the conclusion.

Author Response

Thanks a lot for your comments.  About our respones, please see the attachment. We have rewritten some contents in revised manuscript.

Round 3

Reviewer 1 Report

The authors have updated the article and it looks like more comprehensive. The 'robust' word can be removed from the title.

Author Response

Thank you for you comment.
We have deleted  the " Robust " in the title. We also corrected some grammar mistakes.